# THE COST OF KNOWING: HALLUCINATION QUEST GAME IN RESOURCE-CONSTRAINED MULTI-AGENT SYSTEMS

## ABSTRACT

Current LLM hallucination benchmarks are predominantly static, focusing on factuality while ignoring the computational resources consumed. This creates a distorted view of performance, as costly mitigation strategies can obscure the inherent capabilities of more efficient models. This limitation is especially critical in multi-agent systems (MAS), where resource efficiency and strategic interaction are paramount. To address this gap, we introduce MAS-HQ (**M**ulti-**A**gent **S**ystem **H**allucination **Q**uest Game), a dynamic, game-theoretic framework that evaluates MAS hallucination under strict resource constraints and direct adversarial competition. Within MAS-HQ, agents compete to produce low-hallucination summaries while minimizing resource use. Success is measured by a multi-dimensional metric that explicitly balances factual accuracy against resource penalties, forcing a trade-off between quality and efficiency. We instantiate this competition with Q-Agent, a modular agent architecture designed for strategic play, within a setting that features partial observability to drive tactical decision-making. Our experiments reveal the emergence of diverse winning strategies—some prioritizing high factuality, others superior resource efficiency—and demonstrate adaptive agent behaviors driven by the competitive dynamics. MAS-HQ establishes a principled paradigm for benchmarking hallucination in MAS and provides crucial insights into agent strategies under adversarial, resource-constrained conditions.

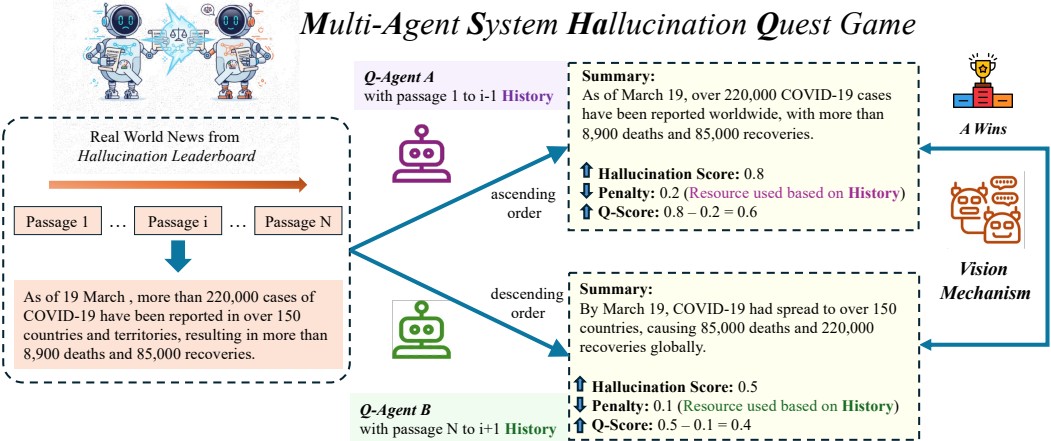

Figure 1: **Overview of the competitive game between two Q-Agents on MAS-HQ.** The original passage list is input in different sequences to Q-Agent A and Q-Agent B. Both Q-Agents devise strategies to generate summaries for each passage using mechanisms such as the vision mechanism. The hallucination scores and resource usage penalties are recorded, leading to the final metric, the Q Score. The agent with the higher Q-Score wins. In the current context, Q-Agent A wins.

# 1 INTRODUCTION

The remarkable advancements in Large Language Models (LLMs) are frequently shadowed by their tendency towards hallucination—generating outputs that appear plausible yet are factually erroneous or unsubstantiated (Macpherson & Platchias, 2013; Huang et al., 2021; Ji et al., 2023; Li et al., 2022). This critical issue is compounded in Multi-Agent Systems (MAS), where interactions between agents can amplify and propagate such inaccuracies (Schmidgall & Moor, 2025). However, progress in this area is hindered by a critical flaw in current evaluation methodologies: they are largely **static**. That is, existing benchmarks and leaderboards (Huang et al., 2025a; Alzahrani et al., 2024; Huang et al., 2025b; Singh et al., 2025) focus almost exclusively on final hallucination scores while ignoring the computational resources consumed to achieve them. This static approach creates a distorted picture, as it allows models to achieve high factuality through costly mitigation strategies—such as extensive API calls or complex reasoning chains—without penalty, obscuring the inherent capabilities of agents under realistic constraints. This oversight is particularly problematic for MAS, where competitive dynamics and stringent operational budgets are paramount, demanding evaluation frameworks that reflect these practical realities.

To address this gap, we introduce MAS-HQ (**M**ulti-**A**gent **S**ystem **H**allucination **Q**uest Game), a novel framework designed to evaluate hallucination in MAS under strict, explicitly defined resource limits and direct adversarial competition. MAS-HQ moves beyond such static tests by placing evaluation in a **dynamic**, game-theoretic setting, which compels agents to strategically manage the trade-off between minimizing hallucination and conserving resources within a competitive environment. The core task within MAS-HQ requires agents to produce low-hallucination summaries of text passages while simultaneously optimizing for minimal resource consumption—measured in terms of token count, API calls, and runtime. This setup is designed to rigorously test an agent's ability to navigate the critical trade-off between factual accuracy and operational efficiency. Consequently, an agent's success is not judged on factuality alone but through a multi-dimensional metric that balances its hallucination score against penalties for resource consumption.

Within this paradigm, we propose Q-Agent, a modular, structured agent architecture designed to navigate MAS-HQ's strategic complexities. Each Q-Agent, with distinct Policy, Summary, Review, and Evaluation modules, competes head-to-head against another Q-Agent. A key feature of MAS-HQ's competitive design is its novel "vision mechanism," introducing partial observability by selectively revealing aspects of an opponent's state (e.g., performance metrics and resource usage) when specific actions like passage review are taken. This turns decisions into strategic gambits: a review may improve summary quality but risks divulging critical information to the adversary, forcing agents to weigh refinement benefits against tactical exposure risks. This fosters a rich environment for emergent strategic behaviors, compelling agents to adapt policies in response to adversarial actions and the evolving game state. An overview of the Q-Agent game under MAS-HQ is shown in Figure 1.

Our extensive experiments within MAS-HQ demonstrate the emergence of diverse winning strategies: some Q-Agents triumph by prioritizing exceptionally high hallucination score (high factual consistency), while others achieve victory through superior resource efficiency, even if their hallucination scores are marginally lower. Further ablation studies validate the framework's robust design, confirming its generality on question-answering tasks, scalability to N-player scenarios, and the necessity of its core competitive mechanisms for inducing these strategic dynamics. These findings underscore the nuanced interplay between factual accuracy, resource management, and adversarial pressure. More broadly, MAS-HQ establishes a principled and practically relevant paradigm for hallucination benchmarking in MAS. It offers crucial insights into agent behavior under conditions of resource scarcity and competitive stress, thereby contributing to the development of more robust, reliable, and strategically adept MAS capable of operating effectively in complex, real-world scenarios.

In summary, our contributions are: (1) **MAS-HQ Framework:** A novel game-theoretic benchmark for evaluating MAS hallucination, emphasizing strict resource constraints and competitive dynamics to ensure fairness and practical relevance in assessing inherent agent capabilities. (2) **Q-Agent and Competitive Paradigm:** A modular agent (Q-Agent) designed for strategic decision-making in MAS-HQ, which facilitates adversarial competition and features a "vision mechanism" to induce adaptive behaviors based on partial observability of opponents. (3) **Empirical Insights into Emergent Strategies:** Extensive experiments demonstrating diverse winning strategies and adaptive agent

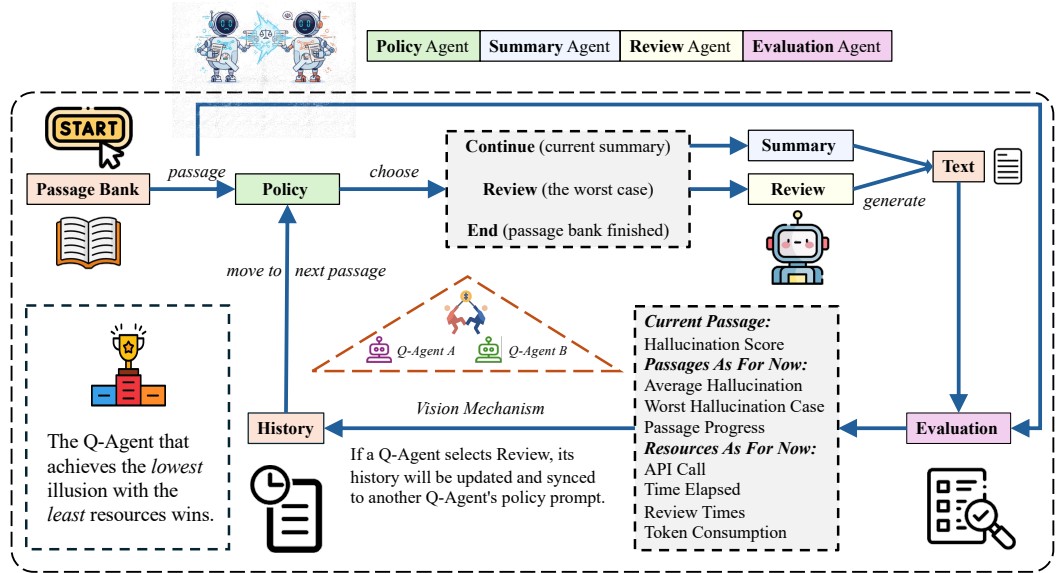

Figure 2: **Overview of the composition of the Q-Agent.** Policy Agent receives both the Q-Agent's own historical information and the opponent's historical information (via the vision mechanism) to determine the next strategy. The text summary generated by either the Summary Agent or the Review Agent is evaluated by the Evaluation Agent, which computes the hallucination score (H-Score) and resource usage. After all steps are completed, the overall Q-Score is calculated, which includes the average hallucination score and resource consumption penalties.

behaviors in MAS-HQ, highlighting the complex interplay of hallucination mitigation, resource management, and adversarial pressure.

## 2 RELATED WORK

**LLM Hallucination** LLM hallucination—the generation of fluent yet factually erroneous content (Macpherson & Platchias, 2013; Huang et al., 2025a)—is a critical challenge. These inaccuracies can be intrinsic (flawed model internals) or extrinsic (misalignment with external knowledge) (Huang et al., 2021; Ji et al., 2023; Li et al., 2022), arising from factors like biased data or uncontrolled inference (Bender et al., 2021; Li et al., 2023; Holtzman et al., 2019). Consequently, a range of mitigation strategies has emerged, from data filtering and retrieval-augmented generation to architectural modifications and decoding algorithms (Abbas et al., 2021; Dai et al., 2023; Gao et al., 2022; Shi et al., 2023; Chuang et al., 2023). However, existing efforts and single-agent benchmarks (Lin et al., 2021; Li et al., 2024; vectara, 2024; Bao et al., 2024; Cheng et al., 2023) predominantly focus on hallucination reduction in isolation, overlooking the crucial resource implications and nuanced behaviors arising from competitive dynamics, a gap our work begins to address by compelling an evaluation of efficiency alongside veracity.

**Multi-Agent Systems** Multi-Agent Systems (MAS) deploy autonomous, LLM-based agents capable of complex planning and interaction (Liu et al., 2025; Guo et al., 2024), showing promise in diverse domains like software development, embodied AI, and scientific discovery (Rasheed et al., 2024; Wang et al., 2024; Huang et al., 2023; Hong et al., 2023; Chen et al., 2023; Yu et al., 2023; Ke et al., 2024; Ni & Buehler, 2024; Xie et al., 2024; Jiang et al., 2024; Wu et al., 2023; Fan et al., 2024; Zhang et al., 2024a;b). As MAS capabilities expand, evaluating their robustness and decision quality under practical constraints becomes paramount. While current MAS benchmarks (Zhu et al., 2025) are valuable for assessing collaboration, they often neglect information fidelity, hallucination propagation, or adversarial interactions under stringent resource limitations. MAS-HQ directly addresses this by introducing a game-theoretic paradigm that compels agents to strategically balance task success, resource management, and hallucination minimization against adversaries, fostering a more realistic and comprehensive evaluation.

## 3 MAS-HQ GAME AND Q-AGENT FRAMEWORK

To evaluate hallucination in resource-constrained, competitive MAS, we introduce MAS-HQ, a dynamic environment compelling agents to balance factual accuracy against resource efficiency and adversarial tactics. This fosters a holistic assessment of agent capabilities and reveals emergent behaviors under pressure, addressing the limitations of static benchmarks. The MAS-HQ evaluation and Q-Agent framework is shown in Figure 2.

### 3.1 MAS-HQ MECHANICS AND Q-AGENT ARCHITECTURE

MAS-HQ's gameplay revolves around a core dual-objective challenge built upon news passages from the Hallucination Leaderboard (vectara, 2024; Bao et al., 2024). Agents must sequentially summarize these passages, simultaneously pursuing high factual consistency (**Minimize Hallucination**) and minimal operational cost (**Optimize Resource Consumption**). Success is quantified by the $Q\text{-}Score$, which elegantly integrates these competing objectives.

This dual-objective design mirrors real-world scenarios where both information quality and generation cost are critical, promoting adaptive and efficient strategies. Integrating hallucination mitigation and resource management, agent success in MAS-HQ is measured by

$$Q\text{-}Score = \frac{1}{N} \sum_{i=1}^{N} \left( \alpha \cdot H\text{-}Score_i - \beta \cdot P_i \right)$$

For each of $N$ passages, $H\text{-}Score_i \in [0, 1]$ is the hallucination score (measures factual consistency, higher is better), $P_i$ is a resource consumption penalty, and $\alpha, \beta > 0$ are fixed weights. This metric rewards an optimal balance between low-hallucination summaries and resource parsimony, discouraging inefficient strategies that overspend for marginal accuracy gains.

**The Q-Agent Architecture.** To navigate this strategic environment, we designed the Q-Agent, a modular architecture comprising four specialized components. The **Policy Agent (PA)** serves as the strategic core, deciding the next action (`continue`, `review`, or `end`) to maximize the final $Q\text{-}Score$ based on its internal state and any opponent intelligence. The **Summarization Agent (SA)** executes the primary task of generating initial summaries. The **Review Agent (RA)** provides a self-correction mechanism, refining existing summaries to improve their $H\text{-}Score$ at the cost of additional resources. Finally, the **Evaluation Agent (EA)** assesses summary quality via an external model (Bao et al., 2024) and tracks all resource consumption, feeding this critical data back to the PA to inform its next strategic move.

### 3.2 ADVERSARIAL DYNAMICS AND STRATEGIC SAFEGUARDS

We instantiate MAS-HQ as a head-to-head competition between two Q-Agents, A and B. To break strategic symmetry and foster dynamic interaction, Agent A processes passages sequentially while Agent B processes them in reverse. The ultimate winner is the agent with the highest final $Q\text{-}Score$.

**The Vision Mechanism.** The competition's strategic depth originates from a novel **vision mechanism** inspired by MOBA games, which induces a state of partial observability. When an agent executes a costly action like 'review', a snapshot of its state ($V_{\text{A},i}^{\text{opponent}}$) is disclosed to its rival. This information leakage allows the opposing agent to dynamically adapt its policy:

$$\text{Choice}_{\text{B},i} = \text{Policy}_{\text{B}}(T_{\text{B},i}, V_{\text{B},i}^{\text{self}}, V_{\text{A},i}^{\text{opponent}}), \quad \text{for} \quad i = 1, \dots, N$$

This transforms every decision into a strategic gambit, forcing agents to constantly weigh the immediate reward of self-improvement against the long-term risk of tactical exposure.

**Operational Safeguards.** To ensure fair competition and deepen strategic complexity, the game is governed by several rules. **Limited Review Cycles ($R$)** cap the number of reviews per passage, encouraging a judicious allocation of refinement resources. **Mandatory Continuation** requires an agent to 'continue' after a 'review', preventing unproductive loops and ensuring forward progress. Lastly, a **Threshold ($T$)-Triggered Guidance** mechanism promotes proactive quality control by encouraging

Table 1: **Main results of Q-Agent Competition across various LLMs.** Within each competition group, Q-Agents are constructed using the same underlying LLM. Performance (H-Score) and resource efficiency (API Calls, Tokens, Review, Time) trade-offs determine the winning agent (Q-Score) for each LLM. The winning agent in each competition is highlighted with a light gray background, and better values for individual metrics are shown in **bold**.

| Q-Agent Competition | Metrics | | | | | | Winner |
|---|---|---|---|---|---|---|---|
| | H-Score ↑ | API Call ↓ | Tokens ↓ | Review ↓ | Time ↓ | Q-Score ↑ | |
| A: GPT-4o-mini | 0.9103 | **2417** | **1.36M** | **791** | **8.83k** | 0.5217 | ✓ |
| B: GPT-4o-mini | **0.9132** | 2438 | 1.44M | 812 | 8.98k | 0.5132 | |
| A: Qwen-Max | **0.9030** | 2304 | **1.48M** | 682 | **13.77k** | 0.5101 | ✓ |
| B: Qwen-Max | 0.8994 | **2264** | 1.52M | **642** | 14.45k | 0.5070 | |
| A: Deepseek-V3 | 0.8860 | 2292 | 1.42M | 666 | 12.26k | 0.4860 | |
| B: Deepseek-V3 | **0.8894** | **2233** | **1.38M** | **607** | **12.06k** | 0.5051 | ✓ |
| A: Gemini-2.0-Flash | **0.9026** | 2262 | 1.56M | 642 | 20.29k | 0.5026 | |
| B: Gemini-2.0-Flash | 0.9016 | **2157** | **1.53M** | **537** | **19.82k** | 0.5273 | ✓ |
| A: Grok-3-beta | **0.9070** | 2376 | 1.37M | 750 | 10.71k | 0.5070 | |
| B: Grok-3-beta | 0.9049 | **2253** | **1.34M** | **627** | **10.15k** | 0.5337 | ✓ |

the PA to 'review' any summary whose hallucination score falls below the predefined threshold $T$. Together, these elements shape a decision-making environment that demands sophisticated reasoning about opponents, resources, and long-term objectives.

## 3.3 Formalization as a Dynamic Game of Imperfect Information

To rigorously ground our framework, we formalize MAS-HQ as a two-player, finite-horizon, general-sum, dynamic game of imperfect information, denoted by $\mathcal{G}$. This game-theoretic perspective is essential for capturing the strategic depth of agent interactions. The core components of $\mathcal{G}$ are defined as follows:

**Players, States, and Actions.** The set of players is $\mathcal{N} = \{A, B\}$. At any discrete time step $i \in \{1, ..., N\}$, the global state of the game $s_i$ contains the complete history of actions and outcomes for both agents. However, the game is one of imperfect information. Each agent $j \in \mathcal{N}$ only has access to a private observation $o_i^j$ which comprises its internal state $V_{j,i}^{\text{self}}$ and, conditionally, a partial signal $\omega_i$ about its opponent's state, $\tilde{V}_{-j,i}^{\text{opponent}}$, which is revealed by the vision mechanism. The action space at each step is $\mathcal{A} = \{\texttt{continue}, \texttt{review}\}$.

**Histories, Beliefs, and Information Structure.** An agent $j$'s local history is the sequence of its past private observations and actions, $h_i^j = (o_1^j, a_1^j, \ldots, o_{i-1}^j, a_{i-1}^j, o_i^j)$. Since $h_i^j$ does not fully reveal the global state $s_i$ (and specifically, the opponent's history $h_i^{-j}$), a rational agent must maintain a *belief state* $b_i^j \in \Delta(\mathcal{H}_i^{-j})$, which is a probability distribution over the set of all possible opponent histories. These beliefs are updated via Bayes' rule whenever new information becomes available. For instance, upon receiving observation $o_i^j$ (which may contain a signal $\omega_i$), the belief is updated from the previous step:

$$b_i^j(h_i^{-j}) = \frac{P(o_i^j|h_{i-1}^j, h_i^{-j}, \pi^{-j*})b_{i-1}^j(h_{i-1}^{-j})}{\sum_{h'^{-j} \in \mathcal{H}_i^{-j}} P(o_i^j|h_{i-1}^j, h'^{-j}, \pi^{-j*})b_{i-1}^j(h_{i-1}')}$$

where the likelihood $P(o_i^j|\cdot)$ depends on the opponent's strategy $\pi^{-j*}$ and the game's stochastic transition dynamics.

**Strategies and Sequential Rationality.** A strategy (or policy) $\pi^j$ for agent $j$ is a complete plan of action that maps each possible history to a probability distribution over actions, $\pi^j : \mathcal{H}^j \to \Delta(\mathcal{A})$. The objective of each agent is to maximize its final expected utility, which is the terminal Q-Score.

Table 2: **Main results investigating the impact of LLM and passage processing order.** In each competition group, Q-Agent A and Q-Agent B are constructed using different LLMs. Q-Agent A retrieves original text from the passage bank in forward order, while Q-Agent B retrieves in reverse order. The winning agent in each competition is highlighted with a light gray background, and better values for individual metrics are shown in **bold**.

| Q-Agent Competition | Metrics | | | | | | Winner |
|---|---|---|---|---|---|---|---|
| | H-Score ↑ | API Call ↓ | Tokens ↓ | Review ↓ | Time ↓ | Q-Score ↑ | |
| A: GPT-4o-mini | **0.9105** | 2401 | 1.34M | 785 | **6.54k** | 0.5401 | |
| B: Grok-3-beta | 0.9035 | **2177** | **1.30M** | **561** | 9.29k | **0.5445** | ✓ |
| A: Grok-3-beta | 0.9036 | **2312** | **1.34M** | **696** | 10.08k | 0.5278 | |
| B: GPT-4o-mini | **0.9092** | 2422 | 1.43M | 806 | **6.78k** | **0.5419** | ✓ |

This can be framed using the Bellman formalism. Let $V^j(h_i^j)$ be the value function for agent $j$ at history $h_i^j$. It represents the maximum expected utility achievable from that point onward. The action-value function is then:

$$Q^j(h_i^j, a_i^j) = \mathbb{E}_{\pi^{-j*}, \mu^*}\left[U^j(\pi) \mid h_i^j, a_i^j\right]$$

Sequential rationality dictates that an agent's strategy must be optimal at every decision point, given its beliefs. Thus, the value of a history is determined by the optimal action:

$$V^j(h_i^j) = \max_{a \in \mathcal{A}} Q^j(h_i^j, a)$$

A rational agent's policy $\pi^{j*}$ will only choose actions that satisfy $a \in \arg\max_{a' \in \mathcal{A}} Q^j(h_i^j, a')$.

**Perfect Bayesian Equilibrium and Strategic Trade-offs.** The canonical solution concept for such games is the Perfect Bayesian Equilibrium (PBE) (Fudenberg & Tirole, 1991). A PBE is a strategy profile $(\pi^{A*}, \pi^{B*})$ and a system of beliefs $\mu^*$ where strategies are sequentially rational for each player given their beliefs, and beliefs are consistent with the strategy profile via Bayesian updates. This game-theoretic formulation is not merely descriptive; it precisely defines the complex optimization problem agents face. The strategic tension arises directly from the PBE structure. An action like 'review' may increase myopic utility by improving a local $H\text{-}Score$, but it simultaneously alters the opponent's belief state $\mu^*$ by revealing information. This leakage can be exploited by a rational opponent, potentially lowering the agent's future expected utility. A rational agent will choose to 'review' passage $k$ only if the expected value from reviewing exceeds that of continuing:

$$\mathbb{E}[V^j(h_{i+1}^j)|a_i^j = \texttt{review}] > \mathbb{E}[V^j(h_{i+1}^j)|a_i^j = \texttt{continue}]$$

This decision hinges on balancing the immediate utility gain against the strategic cost of information leakage, $C_{\text{info}}$. This cost is the expected reduction in future utility resulting from the opponent playing a more informed best response:

$$C_{\text{info}}(h_i^j) \triangleq \mathbb{E}_{\pi^{-j*}(\cdot|b_{\text{prior}}^{-j})}[U^j] - \mathbb{E}_{\pi^{-j*}(\cdot|b_{\text{posterior}}^{-j})}[U^j] > 0$$

While our framework does not compute this equilibrium analytically, its design compels agents to navigate these exact trade-offs, creating a robust testbed for empirically approximating rational, equilibrium behavior under strategic pressure.

## 4 EXPERIMENTS

### 4.1 EXPERIMENTAL SETUP

Our experiments are conducted within MAS-HQ, designed for evaluating MAS on hallucination benchmarks centered around text summarization tasks. MAS-HQ comprises over 1,000 long-form news passages. Factual consistency (or hallucination score), denoted as the $H\text{-}Score$, is assessed

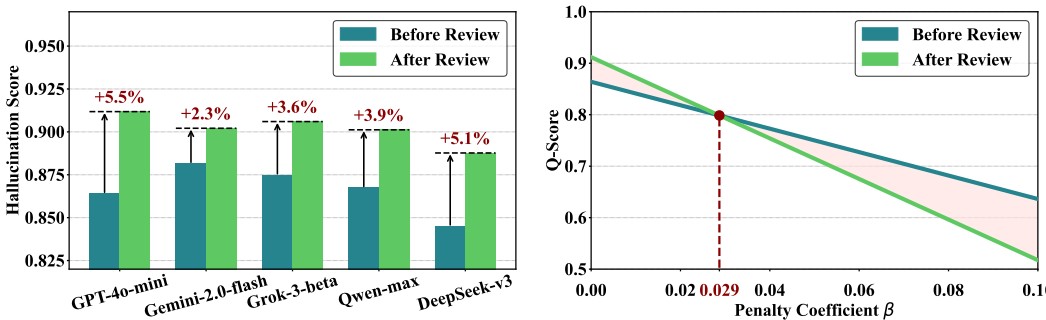

Figure 3: **Ablation study of the Review Agent in Q-Agent.** Each experiment group presents the average results over Q-Agent A and Q-Agent B. *Left:* The Review Agent improves the H-Score and reduces hallucination to varying extents across different Q-Agent models. *Right:* When both Q-Agents are built using GPT-4o-mini, the smaller the penalty coefficient $\beta$, the more significant the improvement in the final Q-Score after review.

using a pre-trained discriminator model (Bao et al., 2024) provided by the hallucination leaderboard (vectara, 2024). A practical challenge encountered was the refusal of some LLMs to summarize passages containing sensitive content (e.g., violence). This behavior affected our Q-Agent's Summarization Agent and Review Agent modules. To address this, we filtered the dataset for each LLM, retaining only those passages it could process. Consequently, the Q-Agent constructed with GPT-4o-mini (Hurst et al., 2024) utilized 808 passages; Grok-3-beta (xAI, 2025), 813 passages; Qwen-Max (Yang et al., 2024), 811 passages; Deepseek-V3 (Liu et al., 2024), 813 passages; and Gemini-2.0-Flash (DeepMind, 2024), 810 passages.

For all experimental configurations, we set the hallucination threshold $T$ for review guidance at 0.85 and the maximum number of review times $R$ per passage at 3. The final $Q\text{-}Score$ is computed with weighting factors $\alpha = 1$ and $\beta = 0.01$. The penalty term $P_i$ for each article $i$ incorporates several factors: the number of API calls, total token consumption (input and output), total review count, and overall runtime. Each component of the penalty is normalized by dividing the current Q-Agent's value by the maximum value observed between itself and its opponent for that component; these normalized values are then summed to form $P_i$. $H\text{-}Score_i$ is directly obtained from the aforementioned discriminator. In all competitive setups, Q-Agent A and Q-Agent B commenced their tasks simultaneously. Q-Agent A processed passages in their original sequence, while Q-Agent B processed them in reverse order. This design choice aimed to mitigate behavioral convergence and encourage the emergence of diverse strategic approaches.

## 4.2 EMERGENT STRATEGIES IN Q-AGENT COMPETITION

We conducted a series of competitions to validate that MAS-HQ elicits the complex, strategic behaviors it was designed to measure. A key critique of existing benchmarks is their narrow focus on accuracy metrics like H-Score, which fails to capture the holistic nature of agent intelligence. Our primary contribution is not a new factuality metric, but rather a new game-theoretic paradigm for evaluation, embodied by the $Q\text{-}Score$. This metric's value lies in its integration of a standard

Table 3: **Scalability to 3-Agent Competition.** Results using GPT-4o-mini demonstrate that the MAS-HQ framework scales to N-player scenarios, enriching the strategic dynamics. Q-Agent A wins via resource efficiency despite having the lowest H-Score.

| Q-Agent Competition | Metrics | | | | | | |
|---|---|---|---|---|---|---|---|
| | H-Score ↑ | API Calls ↓ | Tokens ↓ | Reviews ↓ | Time (s) ↓ | Q-Score ↑ | Winner |
| A: GPT-4o-mini | 0.9108 | **2424** | **1.36M** | **798** | **8.84k** | **0.5214** | ✓ |
| B: GPT-4o-mini | **0.9139** | 2445 | 1.44M | 819 | 8.99k | 0.5139 | |
| C: GPT-4o-mini | 0.9123 | 2434 | 1.40M | 808 | 8.89k | 0.5180 | |

Table 4: **MAS-HQ Generality on SimpleQA Task.** Results show the framework's applicability to question-answering. The H-Score is adapted to exact match accuracy. The core trade-off between performance and resource cost remains, with the winning agent in both cases investing more resources for higher accuracy.

| Q-Agent Competition | Metrics | | | | | | |
|---|---|---|---|---|---|---|---|
| | H-Score ↑ | API Calls ↓ | Tokens ↓ | Reviews ↓ | Time (s) ↓ | Q-Score ↑ | Winner |
| A: GPT-4o | **0.3872** | 11.40k | 0.89M | 2745 | 12.54k | **0.3832** | ✓ |
| B: GPT-4o | 0.3837 | **11.16k** | **0.87M** | **2512** | **12.39k** | 0.3798 | |
| A: GPT-4o-mini | **0.0132** | 11.09k | 0.86M | 2437 | 12.20k | **0.0092** | ✓ |
| B: GPT-4o-mini | 0.0116 | **11.03k** | **0.85M** | **2382** | **12.14k** | 0.0086 | |

factuality score with a multi-faceted resource penalty, compelling a trade-off between accuracy and efficiency. The following experiments demonstrate that winning in MAS-HQ requires navigating this trade-off, revealing strategic nuances that static leaderboards miss.

As shown in our homogeneous and heterogeneous competitions (Table 1 and Table 2), victory is not solely determined by achieving the highest $H$-$Score$. For instance, in the GPT-4o-mini competition, the winning agent had a *worse $H$-$Score$* but triumphed through superior resource efficiency. Conversely, the Qwen-Max winner prioritized a higher $H$-$Score$ at the cost of more API calls. These results directly validate our framework's contribution: it successfully quantifies and reveals the diverse strategic trade-offs agents must make under competitive, resource-constrained conditions. Different LLMs enable distinct paths to victory, underscoring the depth of evaluation MAS-HQ provides beyond simple factuality.

### 4.3 FRAMEWORK VALIDATION AND ABLATION STUDIES

To further validate the robustness, generality, and design choices of our framework, we conducted a series of targeted ablation studies. These experiments demonstrate that MAS-HQ is a versatile paradigm applicable beyond summarization and that its core components are not arbitrary but necessary for inducing meaningful strategic competition.

**Generality and Scalability.** A core design principle of MAS-HQ is modularity. To demonstrate that it is a general framework rather than a task-specific one, we adapted it to a challenging factoid question-answering task using SimpleQA dataset (Wei et al., 2024). The adaptation was straightforward: we replaced the Summarization/Review agents with task-specific QA agents and swapped the $H$-$Score$ metric for exact match accuracy using ChatGPT classifier following the setting in Wei et al. (2024). As shown in Table 4, the framework successfully evaluated the accuracy-resource trade-off in this new domain, confirming its versatility. Furthermore, we tested scalability by extending the competition to three agents (Table 3). The framework scaled effectively, creating a more complex dynamic where the winning agent again prevailed through superior resource management, proving MAS-HQ is a robust paradigm for N-player scenarios.

**Necessity of Core Competitive Mechanisms.** The MAS-HQ game mechanics—the Review Agent, reverse passage order, and the vision mechanism—were deliberately designed as a coherent system to create a non-trivial competitive environment. Ablation studies confirm their necessity. As shown in

Table 5: **Ablation on Passage Order and Vision Mechanism.** The results show that both reverse order processing and the vision mechanism are necessary to break symmetry and induce a dynamic, non-identical competition.

| Setup | Passage Order | Agent A Q-Score | Agent B Q-Score | Outcome |
|---|---|---|---|---|
| **Our Paper's Setup (Vision On)** | **A: Fwd, B: Rev** | **0.5217** | **0.5132** | **Dynamic Competition** |
| Ablation 1 (Vision On) | A: Fwd, B: Fwd | 0.5097 | 0.5097 | Identical (No Asymmetry) |
| Ablation 2 (Vision Off) | A: Fwd, B: Rev | 0.5081 | 0.5081 | Identical (No Interaction) |

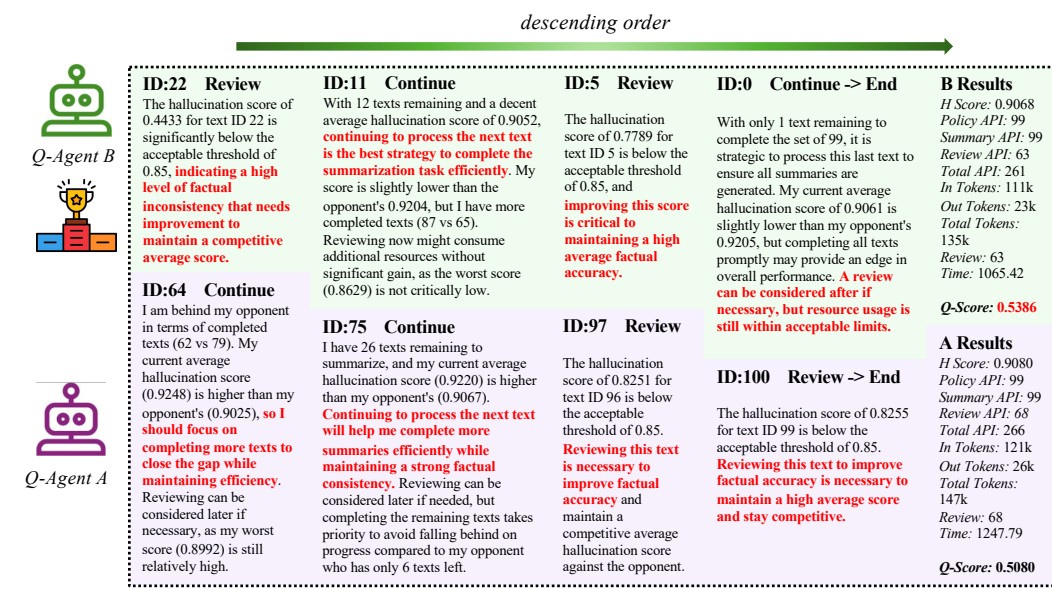

Figure 4: **Case study on a subset of MAS-HQ.** Through vision mechanism, Q-Agent A and Q-Agent B dynamically adjust their strategies based on their own H-Score, resource usage, and comparisons with the opponent, deciding whether to continue generating summaries or perform reviews. In this example, Q-Agent B wins due to its lower resource usage, despite having a slightly lower H-Score.

Figure 3, the **Review Agent** is critical for enabling the core trade-off between improving the $H$-$Score$ and incurring resource costs. The **vision mechanism** and **reverse passage order** are equally vital for breaking strategic symmetry and creating the information asymmetry that drives dynamic interaction. As Table 5 unequivocally show, removing either of these components causes the competition to collapse. Without them, agents lack opponent intelligence or asymmetric starting conditions, leading them to adopt identical strategies and achieve identical scores. This confirms these design choices are not subjective customizations but are fundamental prerequisites for a testbed capable of measuring emergent, adaptive strategies under adversarial pressure.

## 4.4 CASE STUDY

To provide a concrete illustration of emergent agent behaviors, we detail a case study on a 100-passage subset of the game. As depicted in Figure 4, the agents dynamically adapt their strategies based on their own progress, resource usage, and the partial information revealed about their opponent via the vision mechanism. In this instance, Q-Agent B secures a strategic victory with a Q-Score of 0.5386 over Q-Agent A's 0.5080. Notably, Agent B wins despite a marginally lower H-Score (0.9068 vs. 0.9080). Its victory is a direct result of superior resource efficiency, particularly lower token consumption and a shorter runtime. This case perfectly exemplifies an intelligent agent balancing the dual objectives of achieving high factual consistency and maintaining operational efficiency, a nuanced capability that MAS-HQ is uniquely designed to reveal.

## 5 CONCLUSION

MAS-HQ is the first comprehensive game-theoretic framework for rigorously evaluating hallucination in MAS under strict resource limits and challenging adversarial settings. Using the Q-Agent architecture, experiments demonstrated diverse winning strategies that effectively balance accuracy and resource use, alongside adaptive behaviors emerging from competitive interactions. MAS-HQ provides a fairer, more practical, and dynamic benchmarking approach beyond static leaderboards, actively promoting the development of robust, efficient, and resilient MAS.

ETHICS STATEMENT

This work adheres to the ICLR Code of Ethics. Our study does not involve human subjects, sensitive personal data, or experiments that could directly cause harm to individuals or communities. We have taken care to consider issues of fairness, privacy, and security when designing our methods and presenting our results. We are not aware of any potential conflicts of interest, legal compliance issues, or research integrity concerns related to this submission.

REPRODUCIBILITY STATEMENT

We have made every effort to ensure the reproducibility of our results. Details of the model architecture, training procedures, and evaluation protocols are provided in the main text and appendix. Hyperparameters, dataset preprocessing steps, and implementation details are described in the supplementary materials. To further support reproducibility, we upload the source code as supplementary material. These resources should allow other researchers to replicate our findings and build upon our work.

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

## A    THE USE OF LLMS

In the article, we only used LLMs to polish our writing, and did not use them for any other assistance.

## B    DETAILED GAME-THEORETIC FORMALIZATION OF MAS-HQ

In this appendix, we provide a comprehensive formalization of the MAS-HQ framework. We model the system as a two-player, general-sum, finite-horizon, Partially Observable Stochastic Game (POSG), which is a standard and powerful model for multi-agent interactions under uncertainty. This level of detail clarifies the precise mechanics and strategic complexities that the agents must navigate.

A POSG can be formally defined by the tuple $\mathcal{G} = \langle \mathcal{N}, \mathcal{S}, \{\mathcal{A}^j\}_{j \in \mathcal{N}}, \mathcal{T}, \mathcal{R}, \{\Omega^j\}_{j \in \mathcal{N}}, \mathcal{O}, H \rangle$. We define each component in the context of MAS-HQ.

### B.1    CORE COMPONENTS OF THE MAS-HQ GAME

**Players ($\mathcal{N}$):**    The set of players is $\mathcal{N} = \{A, B\}$, representing the two competing Q-Agents.

**State Space ($\mathcal{S}$):**    The global state space $\mathcal{S}$ captures the complete, objective state of the game at any time step. A state $s_i \in \mathcal{S}$ at step $i$ is a composite tuple:

$$s_i = (s_i^A, s_i^B, I_i)$$

where $I_i$ tracks the current passage index for each agent, and $s_i^j$ is the private state of agent $j$, invisible to its opponent. This private state is itself a detailed record of performance and resource expenditure:

$$s_i^j = \langle \{H_k^j\}_{k=1}^N, \{C_k^j\}_{k=1}^N, \mathbf{p}_i^j, \rho_i^j, t_i^j \rangle$$

- $\{H_k^j\}_{k=1}^N$ is the vector of H-Scores for all passages, with entries for unprocessed passages set to a null value.
- $\{C_k^j\}_{k=1}^N$ is the vector of completion statuses for all passages (e.g., not started, summarized, reviewed).
- $\mathbf{p}_i^j$ is the vector of cumulative resource penalties incurred up to step $i$, including token counts, API calls, etc.
- $\rho_i^j$ is the number of remaining review cycles available to agent $j$.
- $t_i^j$ is the cumulative runtime for agent $j$.

**Action Space ($\mathcal{A}$):**    The joint action space is $\mathcal{A} = \mathcal{A}^A \times \mathcal{A}^B$. At any step $i$, each agent $j$ selects an action $a_i^j \in \mathcal{A}^j = \{\texttt{continue}, \texttt{review}_k, \texttt{end}\}$, where $k \in \{1, ..., N\}$ specifies which passage to review. The Policy Agent's decision maps to one of these grounded actions.

**Transition Function ($\mathcal{T}$):**    The transition function $\mathcal{T} : \mathcal{S} \times \mathcal{A} \to \Delta(\mathcal{S})$ defines the dynamics of the game, specifying the probability $P(s_{i+1}|s_i, \mathbf{a}_i)$ of transitioning to state $s_{i+1}$ given the current state $s_i$ and joint action $\mathbf{a}_i = (a_i^A, a_i^B)$. Most transitions are deterministic (e.g., choosing 'continue' increments the passage index). However, stochasticity arises from the 'review' action, where the resulting $H\text{-}Score_{k,\text{new}}^j$ is a random variable conditioned on the Review Agent's capabilities and the passage complexity.

**Reward Function ($\mathcal{R}$):**    The game has a terminal reward structure. For any non-terminal step $i < H$, the immediate reward for each player is zero. The reward function $\mathcal{R} : \mathcal{S} \to \mathbb{R}^{|\mathcal{N}|}$ is defined as:

$$R^j(s_i) = \begin{cases} 0 & \text{if } i < H \\ \frac{1}{N} \sum_{k=1}^N \left( \alpha \cdot H_k^j - \beta \cdot P_k^j(\mathbf{p}_H^j, \mathbf{p}_H^{-j}) \right) & \text{if } i = H \end{cases}$$

where $H$ is the horizon (total number of passages, $N$), and the penalty $P_k^j$ is computed based on the final resource vectors $\mathbf{p}_H^j$ and $\mathbf{p}_H^{-j}$ of both agents, reflecting the normalization step described in the main text.

**Observation Spaces ($\Omega$) and Observation Function ($\mathcal{O}$):**   This is the core of the imperfect information structure. Each agent $j$ has a private observation space $\Omega^j$. The observation function $\mathcal{O} : \mathcal{S} \times \mathcal{A} \to \Delta(\Omega^A \times \Omega^B)$ gives the probability $P(\mathbf{o}_{i+1}|s_{i+1}, \mathbf{a}_i)$ of the agents observing a joint observation $\mathbf{o}_{i+1} = (o^A_{i+1}, o^B_{i+1})$ after a transition. An observation $o^j_i \in \Omega^j$ is defined as:

$$o^j_i = (s^j_i, \omega^j_i)$$

- $s^j_i$ is the agent's own private state, which it always observes.
- $\omega^j_i$ is the signal received from the opponent. The observation function is designed to implement the "vision mechanism":

$$\omega^j_i = \begin{cases} \phi(s^{-j}_i) & \text{if } a^{-j}_{i-1} \in \{\texttt{review}_k\}^N_{k=1} \\ \emptyset & \text{otherwise} \end{cases}$$

where $\phi(s^{-j}_i)$ is a function that extracts a public snapshot of the opponent's state (e.g., their worst H-Score and total tokens used).

## B.2   BELIEFS, POLICIES, AND EQUILIBRIUM

**Histories and Belief States:**   Since each agent cannot observe the full state $s_i$, it must maintain a belief over the possible states of the opponent. An agent $j$'s history is a sequence of its past actions and observations, $h^j_i = (a^j_0, o^j_1, \ldots, a^j_{i-1}, o^j_i)$. A belief state $b^j_i \in \mathcal{B}^j = \Delta(\mathcal{S})$ is a probability distribution over the global state space $\mathcal{S}$, conditioned on the agent's private history $h^j_i$. The belief is updated recursively using the Bayes filter:

$$b^j_i(s') = P(s_i = s'|h^j_i) = \frac{P(o^j_i|s', a^j_{i-1}, b^j_{i-1}) \sum_{s \in \mathcal{S}} P(s'|s, a^j_{i-1}, \pi^{-j*}) b^j_{i-1}(s)}{P(o^j_i|a^j_{i-1}, b^j_{i-1})}$$

where the update depends on the observation function, the transition function, and a model of the opponent's policy $\pi^{-j*}$.

**Policies and Value Functions:**   A policy $\pi^j : \mathcal{B}^j \to \Delta(\mathcal{A}^j)$ maps an agent's belief state to a distribution over its actions. A rational agent seeks a policy that maximizes its expected terminal utility. This can be solved via dynamic programming over the belief space. The value of a belief state for agent $j$ at step $i$ under a policy profile $(\pi^j, \pi^{-j})$ is given by the Bellman equation:

$$V^j_i(b) = \max_{a^j \in \mathcal{A}^j} \left( R^j(b, a^j) + \sum_{o^j \in \Omega^j} P(o^j|b, a^j, \pi^{-j}) V^j_{i+1}(\tau(b, a^j, o^j, \pi^{-j})) \right)$$

where $R^j(b, a^j) = \sum_{s \in \mathcal{S}} b(s) R^j(s, a^j)$ is the expected immediate reward, and $\tau(\cdot)$ is the belief update function.

**Perfect Bayesian Equilibrium (PBE):**   The central solution concept for this game is the PBE. A PBE is a strategy profile $(\pi^{A*}, \pi^{B*})$ and a system of beliefs $\mu^*$ such that:

1. **Sequential Rationality:** For each player $j$, the policy $\pi^{j*}$ must be a best response to $\pi^{-j*}$ at every possible belief state $b \in \mathcal{B}^j$ that can be reached under the equilibrium strategies. That is, $\pi^{j*}$ must satisfy the Bellman optimality equation above.

2. **Belief Consistency:** The beliefs $\mu^*$ must be derived from the strategy profile $(\pi^{A*}, \pi^{B*})$ using Bayesian updating, wherever possible.

The strategic decision to 'review' is thus a comparison between the expected values $Q^j(b, \texttt{review})$ and $Q^j(b, \texttt{continue})$. The 'review' action may increase the immediate components of the final utility (by improving an $H\text{-}Score$) but incurs two costs: (1) a direct resource penalty captured by $\mathcal{T}$ and $\mathcal{R}$, and (2) a strategic information cost, as revealing information through $\mathcal{O}$ allows the opponent to form a more accurate belief $b^{-j}$, leading to a more effective counter-strategy $\pi^{-j*}$, thereby reducing player $j$'s future expected utility. MAS-HQ is designed to create an environment where these complex, interdependent calculations are necessary for victory, thus providing a deep and holistic benchmark of strategic agent intelligence.

# C   MORE ABLATION STUDIES

**Influence of Hyperparameters $T$ and $R$**   Within the Q-Agent framework, the threshold for review guidance $T$ and the maximum number of review times $R$ are critical hyperparameters. $R$ limits the review investment per passage, encouraging broader resource allocation. $T$ influences the Policy Agent's decision to review by providing a recommendation if a passage's $H\text{-}Score$ falls below this threshold (and $R$ is not exceeded), aiming to maintain a baseline level of factual consistency.

**Threshold $T$:** We fixed $R = 3$ and varied $T \in \{0.8, 0.85, 0.9\}$. As shown in Table 6, increasing $T$ generally leads to higher resource consumption. As $T$ rises, more passages are likely to fall below the threshold, triggering more review recommendations and, consequently, actual reviews by the Policy Agent. This was observed as an increase in total resource usage for both Q-Agent A and B. But a higher $T$ does not automatically guarantee a significantly improved $H\text{-}Score$ or final $Q\text{-}Score$. The experiments showed that while resource consumption increased with $T$, the $H\text{-}Score$ did not exhibit a corresponding significant rise and, in some instances, slightly decreased. This led to a marginal decrease in the final $Q\text{-}Score$. This phenomenon could be attributed to two factors: (i) the Policy Agent may still opt against reviewing despite the recommendation if its internal logic deems the current $H\text{-}Score$ sufficient relative to costs, or (ii) excessive reviews on already reasonably good summaries might yield diminishing returns on $H\text{-}Score$ improvement.

**Maximum Review Times $R$:** We fixed $T = 0.85$ and varied $R \in \{2, 3, 4\}$. As shown in Table 7, increasing $R$ beyond a certain point ($R = 3$ in our tests) showed minimal impact on $H\text{-}Score$ and overall resource consumption; token consumption even saw a slight decrease. The final $Q\text{-}Score$ was highest when $R = 3$, suggesting that simply allowing more reviews per article does not compel Policy Agent to utilize them if it deems further reviews unnecessary or inefficient. A limit of $R = 3$ appeared sufficient for the Q-Agents to achieve a good balance, and further increasing $R$ did not lead to proportionally more reviews or better $H\text{-}Scores$, thus avoiding unproductive resource expenditure.

Table 6: **Effect of Hallucination Threshold** ($T$). Results evaluate Q-Agent (built with GPT-4o-mini) performance and resource consumption with a fixed number of allowed reviews ($R = 3$) and varying $T \in \{0.8, 0.85, 0.9\}$. As $T$ increases, resource consumption rises but leads to a slight decrease in H-Score and overall Q-Score.

| Q-Agent Competition | Metrics | | | | | | |
|---|---|---|---|---|---|---|---|
| | H-Score ↑ | API Call ↓ | Tokens ↓ | Reviews ↓ | Time ↓ | Q-Score ↑ | Winner |
| A: $R = 3, T = 0.8$ | 0.9110 | **2401** | **1.33M** | **785** | **7.03k** | **0.5241** | ✓ |
| B: $R = 3, T = 0.8$ | **0.9141** | 2422 | 1.41M | 806 | 7.33k | 0.5141 | |
| A: $R = 3, T = 0.85$ | 0.9103 | **2417** | **1.36M** | **791** | **8.83k** | **0.5217** | ✓ |
| B: $R = 3, T = 0.85$ | **0.9132** | 2438 | 1.44M | 812 | 8.99k | 0.5132 | |
| A: $R = 3, T = 0.9$ | 0.9074 | **2427** | **1.39M** | **807** | **9.10k** | **0.5132** | ✓ |
| B: $R = 3, T = 0.9$ | **0.9113** | 2429 | 1.46M | 809 | 9.20k | 0.5113 | |

Table 7: **Effect of Maximum Allowed Reviews** ($R$). Results evaluate Q-Agent (built with GPT-4o-mini) performance and resource consumption with a fixed hallucination score threshold ($T = 0.85$) and varying $R \in \{2, 3, 4\}$. Increasing $R$ beyond 3 does not significantly alter H-Score or resource consumption, with the highest overall Q-Score observed at $R = 3$.

| Q-Agent Competition | Metrics | | | | | | |
|---|---|---|---|---|---|---|---|
| | H-Score ↑ | API Call ↓ | Tokens ↓ | Reviews ↓ | Time ↓ | Q-Score ↑ | Winner |
| A: $T = 0.85, R = 2$ | 0.9074 | **2414** | 1.39M | **798** | **6.83k** | **0.5139** | ✓ |
| B: $T = 0.85, R = 2$ | **0.9089** | 2423 | **1.24M** | 807 | 7.04k | 0.5089 | |
| A: $T = 0.85, R = 3$ | 0.9103 | **2417** | **1.36M** | **791** | **8.83k** | **0.5217** | ✓ |
| B: $T = 0.85, R = 3$ | **0.9132** | 2438 | 1.45M | 812 | 8.99k | 0.5132 | |
| A: $T = 0.85, R = 4$ | 0.9101 | **2417** | **1.36M** | **799** | **8.15k** | **0.5197** | ✓ |
| B: $T = 0.85, R = 4$ | **0.9112** | 2426 | 1.45M | 808 | 8.32k | 0.5112 | |

**Statistical Robustness.** To address the concern regarding the robustness of our findings, we re-ran the GPT-4o-mini experiment from Table 1 for 100 independent trials. The results, reported in Table 8 with means and standard deviations, confirm the stability and statistical significance of our conclusions. The extremely low standard deviations across all metrics indicate that the strategic trade-offs captured by our framework are highly consistent. The outcome—Q-Agent A winning via resource efficiency—is reproducible, validating that our single-run experiments reliably represent the agents' behaviors.

Table 8: **Statistical Robustness Analysis.** Mean and standard deviation over 100 independent trials of the GPT-4o-mini competition. The low variance across all metrics confirms the stability and reproducibility of our findings.

| Q-Agent Competition | Metrics (Mean ± Std. Dev.) | | | | | |
|---|---|---|---|---|---|---|
| | H-Score ↑ | API Calls ↓ | Tokens ↓ | Reviews ↓ | Time (s) ↓ | Q-Score ↑ |
| A: GPT-4o-mini | 0.9102±.0009 | 2417±5 | 1.36M±.001M | 791±5 | 8.83k±.05k | **0.5216**±.0009 |
| B: GPT-4o-mini | **0.9132**±.0009 | 2438±5 | 1.44M±.001M | 812±5 | 8.98k±.05k | 0.5131±.0009 |

# D    PROMPTS AND OTHER RESULTS

In the appendix, we first present the prompt composition of each module in the Q-Agent. Table 9 shows the detailed results of Table 1 and Table 2 in the main text, with the addition of the changes in $H\text{-}Score$ before and after review, and input tokens consumption and output tokens consumption; Table 10 presents the detailed results of Tables 6 and Table 7 in the main text, also including the changes in $H\text{-}Score$ before and after review.

---

**User Prompt for Q-Agent in MAS-HQ**

**# Policy Agent**
You are part of a Multi-Agent System engaged in a summarization competition against several opponents, where each agent generates multiple candidate summary texts based on the original news article.
The objective is to produce summaries with the lowest possible hallucination (i.e., highest factual consistency, reflected in higher hallucination scores) while minimizing resource usage—including API calls, token consumption, time, and review attempts, all of which are penalized when excessive.
Your ultimate goal is to outperform your competitors by achieving the best balance between factual accuracy and resource efficiency, and to emerge as the final winner.
Your role is to make strategic decisions about what to do next.
Here is your current status:
- MAS ID: {mas_id}
- Completed texts: {completed_texts}/{total_texts}
- Remaining texts: {remaining_texts}
- Current token usage: {token_usage} tokens
- API calls made: {api_calls}
- Review calls made: {review_calls}
- Current average hallucination score: {avg_score}
- Time elapsed: {time_elapsed:.2f} seconds
You need to decide whether to:
- CONTINUE: Process the next text and generate a new summary
- REVIEW: Revisit and improve the text with the highest hallucination rate (lower hallucination score means more hallucination)
- END: Stop processing if all texts have been summarized and there's no need for review
The current text summarization task is at ID {current_id} out of {max_id} texts.
The text with the highest hallucination rate so far is ID {worst_id}, with hallucination score: {worst_score}, where 0 means most hallucinated and 1 means least hallucinated.
{opponent_info}
IMPORTANT: You must output your decision in the following JSON format:

---

```
{
"choice": "continue" or "review" or "end",
```

```
"question_id": current question ID,
"previous_id": ID of the text with highest hallucination rate
}
```

---

# Summary Agent

You are a chat bot answering questions using data. You must stick to the answers provided solely by the text in the passage provided.
You are asked the question 'Provide a concise summary of the following passage, covering the core pieces of information described.'
{passage}
IMPORTANT: You must output your response in the following JSON format:

---

```
{
"summary": "your summary here"
}
```

---

# Review Agent

You are a chat bot answering questions using data. You must stick to the answers provided solely by the text in the passage provided.
You previously summarized the following passage, but your summary contained hallucinations (hallucination score: {score}, where 0 means most hallucinated and 1 means least hallucinated), which means factual inconsistencies occurred.
Original passage: {passage}
Your previous summary: {previous_summary}
Please provide a new, more accurate summary that strictly adheres to the information in the passage. Focus on improving factual consistency and removing any information not present in the original text.
IMPORTANT: You must output your response in the following JSON format:

---

```
{
"summary": "your revised summary here"
}
```

---

Table 9: The detailed data in Table 1 and Table 2 include $H\text{-}Score$ before the review (to the left of the arrow) and after the review (to the right of the arrow), as well as the input tokens and output tokens.

| Q-Agent Competition | Metrics | | | | | | |
|---|---|---|---|---|---|---|---|
| | H-Score ↑ | API Call ↓ | In Tokens ↓ | Out Tokens ↓ | Reviews ↓ | Time ↓ | Q-Score ↑ |
| A: GPT-4o-mini | $0.8606 \rightarrow 0.9103$ | 2417 | 1193792 | 166277 | 791 | 8832.44 | 0.8715 |
| B: GPT-4o-mini | $0.8673 \rightarrow 0.9132$ | 2438 | 1270282 | 178959 | 812 | 8987.41 | 0.5132 |
| A: Qwen-Max | $0.8689 \rightarrow 0.9030$ | 2304 | 1230775 | 254575 | 682 | 13776.31 | 0.5101 |
| B: Qwen-Max | $0.8658 \rightarrow 0.8994$ | 2264 | 1250367 | 270670 | 642 | 14453.96 | 0.5070 |
| A: Deepseek-V3 | $0.8489 \rightarrow 0.8860$ | 2292 | 1219960 | 202441 | 666 | 12267.18 | 0.4860 |
| B: Deepseek-V3 | $0.8504 \rightarrow 0.8894$ | 2233 | 1185813 | 198797 | 607 | 12068.32 | 0.5051 |
| A: Gemini-2.0-flash | $0.8835 \rightarrow 0.9026$ | 2262 | 1173195 | 394719 | 642 | 20295.29 | 0.5026 |
| B: Gemini-2.0-flash | $0.8794 \rightarrow 0.9016$ | 2157 | 1150648 | 379961 | 537 | 19825.70 | 0.5273 |
| A: Grok-3-Beta | $0.8753 \rightarrow 0.9070$ | 2376 | 1180401 | 195116 | 750 | 10719.56 | 0.5070 |
| B: Grok-3-Beta | $0.8743 \rightarrow 0.9049$ | 2253 | 1160176 | 188357 | 627 | 10150.14 | 0.5337 |
| A: GPT-4o-mini | $0.8613 \rightarrow 0.9105$ | 2401 | 1177136 | 165320 | 785 | 6543.14 | 0.5401 |
| B: Grok-3-Beta | $0.8752 \rightarrow 0.9035$ | 2177 | 1121979 | 178422 | 561 | 9294.05 | 0.5445 |
| A: Grok-3-Beta | $0.8730 \rightarrow 0.9036$ | 2312 | 1161825 | 187466 | 696 | 10082.70 | 0.5278 |
| B: GPT-4o-mini | $0.8566 \rightarrow 0.9092$ | 2422 | 1256067 | 178474 | 806 | 6783.37 | 0.5419 |

Then, we present the pseudocode implemented by Q-Agent on MAS-HQ in Algorithm 1.

**Input:** Passages $T_i$, $i = 1$ to $N$; Q-Agents A and B; Hallucination scoring model
**Output:** Global scores $Q^A$, $Q^B$ for both agents
**for** *each agent $\mathcal{A} \in \{Agent\ A, Agent\ B\}$* **do**
 Initialize self-state $V_1^{self}$;
 **for** *each article $T_i$ in order determined by agent (A: $i = 1..N$, B: $i = N..1$)* **do**
  // Policy decision with optional opponent state
  **if** *$\mathcal{A}$ has received $V_i^{opponent}$* **then**
   $choice_i \leftarrow \text{PA}(T_i, V_i^{self}, V_i^{opponent})$;
  **else**
   $choice_i \leftarrow \text{PA}(T_i, V_i^{self})$;
  **end**
  **if** $choice_i = continue$ **then**
   Generate summary $S_i \leftarrow \text{SA}(T_i)$;
  **else if** $choice_i = review$ **then**
   Identify worst summary $S_w$ based on hallucination score;
   Generate revised summary $S_w \leftarrow \text{RA}(T_w, S_w)$;
   // Referee exposes partial state to opponent
   Referee observes $V_i^{self}$ and shares with opponent as $V_{i+1}^{opponent}$;
  **else if** $choice_i = end$ **then**
   Terminate processing;
   **break**;
  **end**
  // Evaluation step
  Compute hallucination score $H_i \leftarrow \text{EA}(T_i, S_i)$;
  Track resource usage: tokens, time, reviews $\rightarrow P_i$;
  // Update self-state
  Update $V_{i+1}^{self}$ with current metrics;
 **end**
 // Compute final global score
 $Q^{\mathcal{A}} = \frac{1}{N} \sum_{i=1}^{N} (\alpha \cdot H_i - \beta \cdot P_i)$;
**end**
**return** $Q^A$, $Q^B$

**Algorithm 1:** MAS-HQ multi-agent evaluation and competition. Each agent sequentially summarizes passages while minimizing hallucination and managing resources. Review actions reveal partial state to opponents, introducing adversarial strategy.

Table 10: The detailed data in Table 6 and Table 7 include $H\text{-}Score$ before the review (to the left of the arrow) and after the review (to the right of the arrow), as well as the input tokens and output tokens.

| Q-Agent Competition | Metrics | | | | | | |
|---|---|---|---|---|---|---|---|
| | H-Score ↑ | API Call ↓ | In Tokens ↓ | Out Tokens ↓ | Reviews ↓ | Time ↓ | Q-Score ↑ |
| A: $R = 3, T = 0.8$ | $0.8653 \to 0.9110$ | 2401 | 1173893 | 164656 | 785 | 7025.22 | 0.5241 |
| B: $R = 3, T = 0.8$ | $0.8626 \to 0.9141$ | 2422 | 1239197 | 175090 | 806 | 7334.55 | 0.5141 |
| A: $R = 3, T = 0.85$ | $0.8606 \to 0.9103$ | 2417 | 1193792 | 166277 | 791 | 8832.44 | 0.5217 |
| B: $R = 3, T = 0.85$ | $0.8673 \to 0.9132$ | 2438 | 1270282 | 178959 | 812 | 8987.41 | 0.5132 |
| A: $R = 3, T = 0.9$ | $0.8621 \to 0.9074$ | 2427 | 1221860 | 169344 | 807 | 9101.14 | 0.5132 |
| B: $R = 3, T = 0.9$ | $0.8554 \to 0.9113$ | 2429 | 1275735 | 179762 | 809 | 9201.04 | 0.5113 |
| A: $R = 2, T = 0.85$ | $0.8598 \to 0.9074$ | 2414 | 1221271 | 174944 | 798 | 6836.23 | 0.5139 |
| B: $R = 2, T = 0.85$ | $0.8588 \to 0.9089$ | 2423 | 1248994 | 178067 | 807 | 7043.92 | 0.5089 |
| A: $R = 3, T = 0.85$ | $0.8606 \to 0.9103$ | 2417 | 1193792 | 166277 | 791 | 8832.44 | 0.5217 |
| B: $R = 3, T = 0.85$ | $0.8673 \to 0.9132$ | 2438 | 1270282 | 178959 | 812 | 8987.41 | 0.5132 |
| A: $R = 4, T = 0.85$ | $0.8620 \to 0.9101$ | 2417 | 1193519 | 162816 | 799 | 8153.87 | 0.5197 |
| B: $R = 4, T = 0.85$ | $0.8599 \to 0.9112$ | 2426 | 1268954 | 176334 | 808 | 8321.54 | 0.5112 |

