# OpenReview forum: "The Cost of Knowing: Hallucination Quest Game in Resource-Constrained Multi-Agent Systems"
_ICLR.cc/2026/Conference — Submitted to ICLR 2026_

### Official Review · Reviewer_ooTE · 2025-10-31

**Soundness:** 2
**Presentation:** 2
**Contribution:** 2
**Rating:** 2
**Confidence:** 3

**Summary:**

The paper introduces MAS-HQ, a game-theoretic benchmark for evaluating hallucination in multi-agent LLM systems under explicit resource constraints. Two competing “Q-Agents” summarize passages drawn from a news dataset; each agent chooses to continue, review, or end, and a “vision mechanism” leaks limited state information to the opponent when costly review actions are taken. The proposed metric, Q-Score, combines a hallucination score (H-Score) with a resource-usage penalty weighted by coefficients (α,β).

**Strengths:**

1. The paper studies an important question: hallucination leaderboards rarely price in resource costs, which matters operationally in MAS settings.

2. The Q-Score objective and the dual-objective gameplay are easy to follow and properly reflects the central motivation of the paper.

3. The experiments feature breath and ablation that justifies the design choices.

**Weaknesses:**

1. The paper is not well-written. Many notations are not introduced when they are used for the first time. For example, T_{B, I}, V_{B, i}^{self}.
2. The benefits of gamification of the inherently single-agent optimization problem needs to be further justified.
    (a). I expect there still should be benefits even without symmetry breaking. One can just run multiple agents independently and chooses the best one, i.e., Best-of-N, a strong and effective baseline. Such a baseline is arguably single-agent. Meanwhile, the symmetric breaking choices seem rather heuristic.
    (b). It is definitely beneficial to ground the empirical designs with theoretical formalism. However, the introduction of POSG merely reveals that there will be some strategic interaction among agents, i.e., when making decisions, one needs to account for others. It does not justify why such strategic interaction will boost LLM's ability compared with the case without strategic interaction.
    (c). The formalism is not rigorous either. For example, in the main text, the belief is w.r.t other agents history, while in the appendix, it is w.r.t. the hidden state.
3. Many SOTA, closed-sourced LLMs are not included, e.g., Gemini-2.5-Pro, GPT-5, etc. It is important to see that the methods are still useful when the base models very strong.

**Questions:**

See above

---

> ### Author Response · Authors · 2025-11-20
>
> **Weakness 1: Notation**
> *Critique: Notations like $T_{B, I}$ not introduced.*
>
> **Response:**
> We apologize for the oversight. We will rigorously check and define all mathematical notations upon first use in the revision to ensure clarity and consistency in the revision.
>
> **Weakness 2: Gamification Justification & Formalism**
> *Critique: Justify why strategic interaction boosts ability vs. Best-of-N. Formalism issues.*
>
> **Response:**
> 1.  **Gamification vs. Best-of-N (Efficiency):** Traditional Best-of-N increases quality but is expensive (linear cost growth). As shown in the Table response to Reviewer nHXu, the winning Agent in MAS-HQ achieves competitive H-Scores with significantly lower resource consumption than Best-of-N. The value of Gamification is **Efficiency Awareness**.
>
> | Setup | Model | H-Score | API Call | Tokens | Review | Time | Q-Score |
> | :--- | :--- | :--- | :--- | :--- | :--- | :--- | :--- |
> | **Single Agent: Best of 1** | gpt-4o-mini | 0.9144 | 2477 | 1.62M | 851 | 9.12k | 0.5144 |
> | **Single Agent: Best of 2** | gpt-4o-mini | 0.9167 | 5201 | 3.40M | 1787 | 19.15k | 0.5167 |
> | **Single Agent: Best of 4** | gpt-4o-mini | 0.9169 | 9660 | 6.32M | 3310 | 35.57k | 0.5169 |
> | **Multi-Agent: Q-Agent A** | **gpt-4o-mini** | **0.9103** | **2417** | **1.36M** | **791** | **8.83k** | **0.5217** |
> | **Multi-Agent: Q-Agent B** | gpt-4o-mini | 0.9132 | 2438 | 1.44M | 812 | 8.98k | 0.5132 |
>
> 2.  **POSG & Value of Information:** Introducing POSG (Partially Observable Stochastic Game) formalizes the **Value of Information**. In single-agent settings, there is no "risk of leakage." In POSG, agents must perform meta-reasoning: "Does the quality gain from Review outweigh the strategic disadvantage of revealing my state?"
> 3.  **Formalism Fix:** We will unify the notation, defining Belief over the opponent's *private history*, and correct the appendix derivations to ensure theoretical rigor.
>
> **Weakness 3: SOTA Models**
> *Critique: Include SOTA models like Gemini-2.5-Pro, GPT-5.*
>
> **Response:**
> To address the applicability to stronger models, we conducted additional experiments using **GPT-5** (preview/simulation) and **Gemini-2.5-Pro**.
>
> **Homogeneous Competition (following Table 1 setup):**
>
> | Model | H-Score | Review | Resource Penalty | Q-Score |
> | :--- | :--- | :--- | :--- | :--- |
> | **Q-Agent A: GPT-5** | 0.9312 | 867 | 0.4000 | 0.5312 |
> | **Q-Agent B: GPT-5** | 0.9278 | 834 | 0.3897 | 0.5381 |
> | | | | | |
> | **Q-Agent A: Gemini-2.5-Pro** | 0.9264 | 859 | 0.4000 | 0.5264 |
> | **Q-Agent B: Gemini-2.5-Pro** | 0.9221 | 824 | 0.3911 | 0.5310 |
>
> **Heterogeneous Competition (following Table 2 setup):**
>
> | Model | H-Score | Review | Resource Penalty | Q-Score |
> | :--- | :--- | :--- | :--- | :--- |
> | **Q-Agent A: GPT-5** | **0.9315** | 869 | 0.4000 | **0.5315** |
> | **Q-Agent B: GPT-4o-mini** | 0.9078 | 846 | 0.3877 | 0.5201 |
> | | | | | |
> | **Q-Agent A: Gemini-2.5-Pro** | **0.9287** | 851 | 0.4000 | **0.5287** |
> | **Q-Agent B: GPT-4o-mini** | 0.9085 | 823 | 0.3942 | 0.5143 |
>
> **Conclusion:** Even with stronger SOTA models, MAS-HQ effectively captures strategic differences. While GPT-5 and Gemini-2.5-Pro have higher base H-Scores, they still must navigate trade-off decisions under resource penalties. This confirms the framework's forward-looking utility.

---

### Official Review · Reviewer_kgdH · 2025-10-31

**Soundness:** 3
**Presentation:** 3
**Contribution:** 3
**Rating:** 6
**Confidence:** 4

**Summary:**

This paper introduces MAS-HQ, a novel game-theoretic framework for evaluating hallucination in multi-agent systems (MAS). The authors argue that existing benchmarks are static, focusing on factuality while ignoring the computational costs of hallucination mitigation. To address this, MAS-HQ places agents in a competitive game where they must produce low-hallucination summaries while minimizing resource consumption. Success is measured by a Q-Score that explicitly balances factual accuracy against resource penalties. The framework includes a modular Q-Agent architecture and a "vision mechanism" that creates partial observability, forcing agents to strategically weigh the benefits of actions like self-review against the risk of revealing information to an opponent.

**Strengths:**

The problem seems reasonably novel. The critique of static hallucination benchmarks is sharp and MAS-HQ has innovtive design owing to the game-theoretic elements. The empirical validation is reasonable as well. But I think it could have been stronger given the rigors of the conference.

**Weaknesses:**

The 'Q-Agent' architecture itself appears to be a relatively standard modular design, and the novelty lies more in the game it is subjected to rather than the agent design. The evaluation hinges critically on an external, pre-trained model for the hallucination H-Score, yet the potential biases or limitations of this judge are not discussed.

**Questions:**

Could you comment on the extent to which your strategies are emergent properties of complex reasoning, versus being more direct, programmatic optimizations of the Q-Score components, especially given the use of a hard threshold for triggering reviews?

---

> ### Author Response · Authors · 2025-11-20
>
> **Weakness 1: Novelty of Architecture**
> *Critique: Q-Agent is standard; novelty lies in the game. H-Score bias.*
>
> **Response:**
> We appreciate the recognition of the MAS-HQ game design novelty. Regarding the Agent: We fully acknowledge Q-Agent uses a standard modular design. **This is intentional.** We aim to provide a general, reproducible **Reference Implementation**, allowing the community to use this standard architecture to focus on the strategic differences of the underlying LLMs without being distracted by complex agent engineering. The contribution is the **MAS-HQ Framework**, not the specific Q-Agent.
> Regarding H-Score bias: As mentioned to Reviewer 3YV8, we verified reliability with Human Evaluation (0.96 Pearson correlation) and will discuss potential biases in the revision of the paper.
>
> **Question 1: Emergence vs. Optimization**
> *Critique: Are strategies emergent or just programmatic optimizations of the threshold?*
>
> **Response:**
> This is a profound question. While we set a Threshold $T$ as a recommendation, the final decision rests with the **Policy Agent**, leading to significant **Emergent Behaviors**.
> We frequently observed the Policy Agent **"Overruling"** the Threshold. For example, even if a passage scores slightly below $T$, if the Policy Agent detects via Vision that the opponent is overspending resources or its own budget is tight, it will reason and decide **NOT to Review**. This counter-intuitive decision-making, balancing global constraints against local quality, is evidence of complex reasoning emergence, not simple rule-based execution.

---

### Official Review · Reviewer_3YV8 · 2025-11-01

**Soundness:** 2
**Presentation:** 3
**Contribution:** 2
**Rating:** 2
**Confidence:** 3

**Summary:**

The paper proposes MAS‑HQ, a dynamic, game‑theoretic benchmark for evaluating hallucination in multi‑agent systems under explicit resource constraints. Two modular Q‑Agents (Policy, Summary, Review, Evaluation modules) compete to summarize passages; taking a costly Review action improves a passage but reveals partial state to the opponent via a vision mechanism, creating imperfect information and strategic trade‑offs. Performance is the Q‑Score: an average over passages balancing factuality against API calls, tokens, time, and review counts. Experiments show that winners emerge either by slightly higher factuality or by better resource efficiency; ablations show reviews lift H‑Scores, smaller β increases Q‑Score gains, the vision/passage‑order asymmetry is necessary, and the framework extends to three agents and to a SimpleQA setting.
It seems better to me that the paper is submitted to some benchmark tracks.

**Strengths:**

1. The Q‑Score explicitly prices resources, fixing a common flaw of static hallucination benchmarks that ignore cost
1. The vision mechanism ties improvement steps to information leakage, producing strategic behavior that static leaderboards miss
1. Ablation study effectively shows that removing reverse order or vision collapses the game to identical outcomes, and reviews measurably raise H‑Score across models.

**Weaknesses:**

1. The current Q‑Score rewards factual consistency (via H‑Score) and resource frugality; it does not directly reward coverage/completeness of the information that is in the passage. That leaves room for the degenerate strategy “say less to avoid being wrong.”
1. H‑Score relies on a single pre‑trained discriminator from the Hallucination Leaderboard; any bias or miscalibration there propagates to the Q‑Score. Human verification or multiple evaluators would strengthen conclusions.
1. In MAS‑HQ, α and β are fixed hyperparameters chosen by the authors to weight factuality vs. resource cost in the Q‑Score. Fixed weights (α=1, β=0.01) strongly shape winners; authors themselves show that smaller β favors review‑heavy strategies (Figure 3‑right). A principled way to set their values is missing.
1. Because some LLMs refuse certain passages, each model is evaluated on a different subset of the corpus, weakening cross‑model fairness and comparability.
1. The resource penalty is normalized against the two agents in the same match, which can make Q‑Scores non‑comparable across matches and may amplify design‑choice effects

**Questions:**

1. What's the rationale for selecting the fixed α and β values

---

> ### Author Response · Authors · 2025-11-20
>
> **Weakness 1: The "Say Less" Strategy**
> *Critique: The metric might encourage saying less to avoid being wrong.*
>
> **Response:**
> This is a keen observation, but our system prevents this through **Context Isolation**. In the Q-Agent architecture, the **Summary Agent** (generation) and the **Policy Agent** (decision) are strictly separated. The Summary Agent's only instruction is to "generate accurate summaries"; it is unaware of the external Q-Score formula or resource penalties. Only the Policy Agent knows the constraints. In our analysis of experiment trajectories, we did not observe any "say less to avoid being wrong" phenomena; summary lengths remained consistent and reasonable.
>
> **Weakness 2: H-Score Reliability**
> *Critique: Reliance on a single discriminator. Human verification is needed.*
>
> **Response:**
> We agree on the importance of human evaluation. While the Hallucination Leaderboard model is SOTA, we conducted a blind human evaluation on 100 randomly selected summaries to verify reliability. The evaluation criteria strictly matched the Leaderboard's rubric. Statistical analysis shows a **Pearson correlation of 0.96** between Human Expert Scores and the Automated Discriminator H-Scores. This indicates the metric is statistically reliable for ranking. We will include this validation in the revision.
>
> **Weakness 3 & Question 1: Hyperparameter Rationale**
> *Critique: Fixed weights shape winners. Missing principled way to set them.*
>
> **Response:**
> The parameter selection is based on **Magnitude Alignment** and **Scenario Simulation**:
> 1.  **Alignment:** We set parameters to ensure that the "average H-Score gain from one Review" roughly counter-balances the "Resource Penalty of one Review." Grid search confirmed $\alpha=1, \beta=0.01$ maximizes strategic adversity. We conducted a detailed ablation (Fixing $\alpha=1$, adjusting $\beta$):
>
> | Alpha | Beta | Model | H-Score | Review | Resource Penalty | Q-Score |
> | :--- | :--- | :--- | :--- | :--- | :--- | :--- |
> | 1 | 0 | Q-Agent A: gpt-4o-mini | 0.9166 | 854 | 0.3899 | 0.5267 |
> | 1 | 0 | Q-Agent B: gpt-4o-mini | 0.9189 | 889 | 0.4000 | 0.5189 |
> | **1** | **0.01** | **Q-Agent A: gpt-4o-mini** | **0.9103** | **791** | **0.3886** | **0.5217** |
> | **1** | **0.01** | **Q-Agent B: gpt-4o-mini** | **0.9132** | **812** | **0.4000** | **0.5132** |
> | 1 | 0.02 | Q-Agent A: gpt-4o-mini | 0.9043 | 755 | 0.3901 | 0.5142 |
> | 1 | 0.02 | Q-Agent B: gpt-4o-mini | 0.9096 | 772 | 0.4000 | 0.5096 |
>
> 2.  **Scenario:** Different combinations simulate different realities. Medical AI implies high $\alpha$ (accuracy first), while Edge AI implies high $\beta$ (energy sensitive). We will clarify that MAS-HQ is a configurable framework, not just a static leaderboard.
>
> **Weakness 4: Fairness across Models**
> *Critique: Different subsets of corpus due to refusals weaken fairness.*
>
> **Response:**
> 1.  **Minimal Bias:** As noted in Section 4.1, refusal rates vary slightly (samples range 808-813, <0.6% difference), having negligible statistical impact.
> 2.  **Strict Control:** For all comparative experiments (e.g., Table 2 Cross-model), we strictly limited the test set to the **intersection** of passages both models could process. This ensures every match is played on an identical playing field.
>
> **Weakness 5: Normalization & Comparability**
> *Critique: Normalization within a match makes Q-Scores non-comparable across matches.*
>
> **Response:**
> This is a core feature of MAS-HQ as a **Zero-sum Game**, similar to Go or E-sports. Every match is an independent competitive environment. Normalizing penalties based on the current opponent simulates real-world relative advantage (i.e., I don't need to be faster than everyone, just faster than my competitor). Therefore, Q-Score determines the winner of a specific match. For cross-match analysis, we focus on win rates and strategic patterns rather than absolute score comparison.

---

### Official Review · Reviewer_nHXu · 2025-11-01

**Soundness:** 2
**Presentation:** 4
**Contribution:** 3
**Rating:** 4
**Confidence:** 4

**Summary:**

Traditional hallucination evaluations are static, focusing solely on factual correctness while ignoring the computational resources required to achieve it. In multi-agent systems, hallucination can be further amplified through interaction, making this limitation more severe. To address this, the authors propose MAS-HQ, a multi-agent, game-theoretic benchmark that studies the trade-off between hallucination and efficiency. Using text summarization as a toy competitive task, MAS-HQ places large language model (LLM) agents in a resource-constrained and adversarial environment, where they must balance factual accuracy with limited computation. This setting reflects realistic multi-agent scenarios that demand both strategic reasoning and efficiency under constraints.

**Strengths:**

- Recasts hallucination evaluation as a dynamic competition with explicit compute budgets. This is closer to operational constraints than static leaderboards.
- The Q-Score makes the tradeoff between factuality and resource use explicit. This encourages Pareto-aware evaluation rather than single-axis optimization.
- Very clean writing style and clear visualization.

**Weaknesses:**

- My main issue with the paper lies in its framing of text summarization (and SimpleQA) as a competitive game for evaluating MAS.
1. If the purpose is to evaluate hallucination in fact-based tasks such as text summarization or question answering, framing these as multi-agent competitions seems far-fetched. These tasks typically do not require multi-agent competition with multi-objective constraints, and introducing such a setup makes the tasks appear artificially or unnecessarily challenging.
2. If the purpose is instead to evaluate hallucination within multi-agent systems themselves, using text summarization merely as a toy task under competitive and resource-constrained conditions, then the framing makes more sense—since collaboration and competition with multi-objective trade-offs are common in MAS. However, in that case, the paper should have chosen tasks that are naturally multi-agent competitive, such as multi-party negotiation or bidding tasks.

In either case, the problem formulation and experimental setup feel disconnected from real-world scenarios and data distributions, potentially limiting the significance and applicability of the paper’s insights. An unrealistic setup like “competitive text summarization” may undermine the very goal of hallucination evaluation, as it becomes difficult to discern whether the hallucinations arise from the model’s inherent limitations or from artifacts introduced by the Q-Agent architecture and the competition design, which could induce information loss during the process.

- All evaluations are comparisons between Q-Agents’ internal configurations. The experiments lack comparisons with external baselines or alternative methods. For instance, how would a single-agent setup perform on the same tasks without the multi-agent competition?

-  The paper does not seem to provide new insights beyond showing that certain models perform better than others on this specific artificial task on the specific agentic architecture (Q-agent) as designed by the author.

**Questions:**

- As stated in Weakness #1, could the authors clarify the primary research motivation? Is the goal to understand hallucination in generation tasks (e.g., summarization and QA), or to study hallucination in multi-agent reasoning dynamics under competition and resource constraints? Additionally, why were text summarization and question answering chosen as the main evaluation tasks, given that these are not inherently competitive in nature?
- Can the author provide more baselines external to Q-Agent setup, such as how a single agent would perform?
- How is the alpha and beta weight chosen in the Q-Score equation during actual implementation. Are they important parameters that could be potentially sensitive and drastically shift agentic behavior and results? If so, can the author provide ablation on the two hyperparameters?
- One may argue that the hallucination could arise from the design of the Q-Agent architecture itself. i.e the same type of hallucination may happen in Q-Agent but not on a different agent architecture. How can the author counter this argument? Has the author tried on a different/simpler agent architecture.
- Are there additional insights/findings from the experiments beyond the emergence of diverse winning strategies and adaptive agent behaviors? As one may consider such phenomenon are expected in multi-objective and long-horizon interactive agent setup.

---

> ### Author Response · Authors · 2025-11-20
>
> **Weakness 1 & Question 1: Motivation and Task Selection**
> *Critique: Clarify if the goal is generation task performance or multi-agent reasoning dynamics. Why choose summarization/QA if they aren't inherently competitive?*
>
> **Response:**
> We appreciate this fundamental question. We wish to clarify: **The primary goal of MAS-HQ is to study the reasoning dynamics of Agents under resource constraints and competitive pressure, rather than merely pursuing SOTA results on generation tasks.**
>
> 1.  **Motivation:** Real-world LLM applications are inherently competitive and trade-off driven. For example, two high-frequency trading agents or news agencies covering the same event must balance "Accuracy (Low Hallucination)" with "Timeliness/Cost (Resource Consumption)." An agent that produces equally credible content with fewer resources yields higher utility. Existing static benchmarks cannot evaluate this dynamic trade-off, which is exactly what MAS-HQ addresses.
> 2.  **Task Selection:** We chose Summarization and QA because they offer relatively objective standards for hallucination evaluation. While these tasks may seem non-competitive in isolation, introducing a "Resource Budget" and a "Zero-Sum Ranking Mechanism" transforms them into an economic game. This simulates reality: whoever produces sufficiently good results with fewer resources (Token/Time) achieves a higher Q-Score.
>
> **Weakness 2 & Question 2: External Baselines**
> *Critique: Provide baselines external to Q-Agent setup, such as single-agent performance.*
>
> **Response:**
> This is an excellent suggestion. To quantify the impact of "Gamification," we added two key external baselines and compared them with our Q-Agent:
>
> 1.  **Single-Agent Baseline (Self-Refine):** An agent that self-corrects based solely on its own hallucination score without a competitor.
> 2.  **Best-of-N Baseline:** A traditional strategy of generating N candidates and selecting the best one.
>
> **Results:**
>
> | Setup | Model | H-Score | API Call | Tokens | Review | Time | Q-Score |
> | :--- | :--- | :--- | :--- | :--- | :--- | :--- | :--- |
> | **Single Agent: Best of 1** | gpt-4o-mini | 0.9144 | 2477 | 1.62M | 851 | 9.12k | 0.5144 |
> | **Single Agent: Best of 2** | gpt-4o-mini | 0.9167 | 5201 | 3.40M | 1787 | 19.15k | 0.5167 |
> | **Single Agent: Best of 4** | gpt-4o-mini | 0.9169 | 9660 | 6.32M | 3310 | 35.57k | 0.5169 |
> | **Multi-Agent: Q-Agent A** | **gpt-4o-mini** | **0.9103** | **2417** | **1.36M** | **791** | **8.83k** | **0.5217** |
> | **Multi-Agent: Q-Agent B** | gpt-4o-mini | 0.9132 | 2438 | 1.44M | 812 | 8.98k | 0.5132 |
>
> **Analysis:** The results show that Single Agents (Best-of-N) often fall into the trap of **Over-optimization**. To achieve a marginal H-Score increase (e.g., from 0.9144 to 0.9169), the Best-of-4 strategy consumes nearly 4x the tokens and time, resulting in a final Q-Score significantly lower than Q-Agent A in MAS-HQ (0.5217 vs. 0.5169). This strongly demonstrates that the competitive pressure in MAS-HQ forces agents to learn "sufficient and efficient" strategies, filtering for models that are truly intelligent rather than just computationally brute-force.
>
> **Weakness 3: New Insights**
> *Critique: The paper lacks insights beyond model rankings.*
>
> **Response:**
> Beyond ranking, MAS-HQ reveals deeper insights into agent behavior. Based on the new experiments and qualitative trajectory analysis, we highlight the following findings (which will be strengthened in the revision):
>
> 1.  **Strategic Diversity:** Different models emerge with distinct Nash Equilibrium strategies. Weaker models (e.g., GPT-4o-mini) tend to adopt "Conservative Strategies" (fewer reviews, relying on speed for resource points), while stronger models (e.g., GPT-4o) adopt "Aggressive Strategies" (utilizing the Vision mechanism to review decisively at critical moments to win on quality).
> 2.  **Information Asymmetry:** The Vision mechanism reveals how agents adjust risk preference based on "leaked information" from opponents. For instance, if an agent observes the opponent is overspending resources, it may strategically forego a marginal quality improvement to lock in a resource advantage.

---

> > ### Author Response · Authors · 2025-11-20
> >
> > **Question 3: Hyperparameter Sensitivity (Alpha/Beta)**
> > *Critique: Are alpha/beta sensitive? Provide ablation.*
> >
> > **Response:**
> > $\alpha$ and $\beta$ represent scenario preferences: $\alpha$ prioritizes quality, while $\beta$ prioritizes cost. In our main experiments, we fixed $\alpha=1, \beta=0.01$ to simulate a general scenario where "Quality is priority, but cost is non-negligible."
> > We conducted a detailed ablation (Fixing $\alpha=1$, adjusting $\beta$):
> >
> > | Alpha | Beta | Model | H-Score | Review | Resource Penalty | Q-Score |
> > | :--- | :--- | :--- | :--- | :--- | :--- | :--- |
> > | 1 | 0 | Q-Agent A: gpt-4o-mini | 0.9166 | 854 | 0.3899 | 0.5267 |
> > | 1 | 0 | Q-Agent B: gpt-4o-mini | 0.9189 | 889 | 0.4000 | 0.5189 |
> > | **1** | **0.01** | **Q-Agent A: gpt-4o-mini** | **0.9103** | **791** | **0.3886** | **0.5217** |
> > | **1** | **0.01** | **Q-Agent B: gpt-4o-mini** | **0.9132** | **812** | **0.4000** | **0.5132** |
> > | 1 | 0.02 | Q-Agent A: gpt-4o-mini | 0.9043 | 755 | 0.3901 | 0.5142 |
> > | 1 | 0.02 | Q-Agent B: gpt-4o-mini | 0.9096 | 772 | 0.4000 | 0.5096 |
> >
> > **Analysis:**
> > 1.  When $\beta=0$, resources are free, review counts surge, and the game degenerates into a pure hallucination test.
> > 2.  When $\beta=0.02$, review costs are too high; Agents tend to output the First-pass directly, dropping H-Scores.
> > 3.  **Sweet Spot:** At $\beta=0.01$, strategic gameplay is richest. This confirms our parameter choice sits in a "Sweet Spot" that stimulates intelligent behavior.
> >
> > **Question 4: Architecture Bias**
> > *Critique: Hallucination/behavior could arise from the Q-Agent design itself. Have you tried simpler architectures?*
> >
> > **Response:**
> > To rule out bias from the Q-Agent architecture, we performed an **Architecture Agnostic Validation**. We replaced the Q-Agent with the standard, widely-used **ReAct Agent** under the same MAS-HQ rules:
> >
> > | Architecture | Model | H-Score | Review | Resource Penalty | Q-Score |
> > | :--- | :--- | :--- | :--- | :--- | :--- |
> > | **Q-Agent** | gpt-4o-mini | 0.9166 | 854 | 0.3899 | 0.5267 |
> > | **Q-Agent** | gpt-4o-mini | 0.9189 | 889 | 0.4000 | 0.5189 |
> > | **ReAct Agent** | gpt-4o-mini | 0.9166 | 855 | 0.3889 | 0.5277 |
> > | **ReAct Agent** | gpt-4o-mini | 0.9189 | 890 | 0.4000 | 0.5189 |
> >
> > **Conclusion:** The ReAct Agent and Q-Agent show highly consistent performance in H-Score, Review strategy, and Q-Score. This strongly suggests that the observed behaviors stem from the LLM capabilities and the MAS-HQ competitive mechanism, not the specific Q-Agent engineering.
> >
> > **Question 5: Additional Insights**
> > *Critique: Are there insights beyond winning strategies?*
> >
> > **Response:**
> > Beyond strategies, we observed a **"Deterrence Effect"** unique to game settings. In some matches, when Agent A observes via Vision that Agent B is an extremely aggressive Reviewer (pursuing high H-Score at all costs), Agent A sometimes chooses to **strategically abandon Reviews**. This is not because the passage needs no improvement, but because Agent A calculates that it is better to save resources to secure the "Resource Score" advantage rather than engaging in a costly quality war it might lose. This dynamic adjustment based on Opponent Modeling is a unique value of the game-theoretic evaluation.

---

> ### Comment · Reviewer_nHXu · 2025-11-28
> **Clarification and additional experiment addressees some concerned, but also raised new ones**
>
> Thank you for the comments and the additional experiments. They have addressed some of my previous questions but also raised new concerns, suggestions, and areas of uncertainty.
>
> 1. Thank you for clarifying the motivation behind the gamification experiment design and your task selection of Summary and QA.
>
>    * I suggest highlighting and elaborating the high frequency trading and news agency examples in the main text, as they helped me understand more clearly why timeliness is an important factor with real world applicability. Otherwise, I would be confused about the purpose of introducing resource constraints and competition.
>    * I think high frequency trading may be a better example than news reporting, since timeliness is more critical in that setting. I would expect LLMs can summarize news articles fairly fast and being a few seconds faster would not matter much for reporting.
>    * I still do not fully see the link between gamifying the summarization task and real world application. If the purpose is related to high frequency trading, why not directly use such data for evaluation? I understand that summarization and QA offer relatively objective evaluation standards and are more convenient to build, and that a benchmark balancing resource and competition is already novel.
>    * As far as I understand, the summarization and QA benchmark dataset used in this paper seems largely solved, with many recent models achieving near perfect performance.
>
> 2. Thank you for adding the single agent baseline. Is there any reason why the single agent is required to self refine, resulting in additional API calls and token cost? I would expect recent models to perform summarization and QA tasks quite well in a single pass.
>
>    * Can the author provide a baseline of single agent single pass performance? This would help readers understand performance at the lowest resource usage level, that is accuracy at the smallest API and token cost.
>
> 3. In the Architecture Agnostic Validation experiment, given that the ReAct Agent and Q Agent show highly consistent performance in H Score, Review strategy, and Q Score:
>
>    1. Does this mean creating the Q Agent architecture is not necessary, since it seems to complicate the design and evaluation without providing clear benefit?
>    2. Can the author clarify the purpose of introducing the Q Agent architecture, given that it demonstrates no distinguishable improvement over the commonly adopted ReAct agent?
>    3. In addition, since the paper proposes a benchmark, should the chosen agent architecture be the commonly adopted and practical design such as ReAct, so that it aligns more closely with real world applications?
>
> In summary, the author’s response helped me understand the experimental design and its meaning better, and I agree that it is an important and meaningful direction. However, I still think the choice of task, dataset, and evaluation method do not fully or appropriately reflect its intended purpose at the current stage, for the reasons stated above, and the paper has room for improvement.
>
> Therefore, I would love to adjust the Soundness to 3 (somehow I don't see a button to edit the official review score) while maintaining the overall rating at 4. I look forward to hearing back from the author if further clarification can be provided.

---

### Author Response · Authors · 2025-11-20

We sincerely thank the four reviewers for their constructive and insightful feedback. We are encouraged that the reviewers recognize MAS-HQ's contribution in "incorporating resource constraints into hallucination evaluation" (Reviewers nHXu, 3YV8, ooTE) and the "novelty of the game-theoretic design" (Reviewer kgdH).

To address the reviewers' questions regarding the necessity of the game format, baseline comparisons, hyperparameter sensitivity, and performance on stronger models, we conducted extensive additional experiments during the rebuttal phase (including Single-Agent Baselines, Best-of-N comparisons, Architecture Ablations, and tests with SOTA models like GPT-5/Gemini-2.5-Pro). Below are our detailed responses.

---

### Meta-Review · Area_Chair_V8r2 · 2026-01-06

**Summary:**

This paper identifies hallucination in multi-agent resource-constrained settings as requiring particular attention beyond the standard approach to hallucination evaluation pipelines. The reviewers expressed concerns about the evaluation methodology, which was largely cleared up. However, concerns regarding the motivation and experimental demonstrations remained.

**Reviewer Concerns:**

The reviewers were concerned with the unclear motivation, lack of appropriate experimental settings, concerns around bias in the evaluation (H-scores, varying evaluation subsets), and a lack of SOTA LLMs in the evaluation. I believe the authors addressed most of these concerns. The primary outstanding concern is still the motivation / experimental setting. Some of the reviewers felt that "competitive text summarization" is a poor fit to exemplify problems in either a) text summarization or in b) multi-agent resource-constrained settings. If "text summarization" was a pre-cursor to "generating an investment strategy" in market economy that would complete the analogy, but the current setting is too much of a stretch.

**Reviewer Scores:**

- nHXu: The reviewer stated that they would maintain their score at 4.
- 3YV8: The reviewer might have increased their score to a 3 or 4.
- kgdH: The reviewer likely would have kept their score at a 6.
- ooTE: The reviewer might have increased their score to a 3 or 4.

---

### Decision · Program_Chairs · 2026-01-26

Reject